# Reaction-Based Amine and Alcohol Gases Detection with Triazine Ionic Liquid Materials

**DOI:** 10.3390/molecules25010104

**Published:** 2019-12-27

**Authors:** Hsin-Yi Li, Yen-Ho Chu

**Affiliations:** Department of Chemistry and Biochemistry, National Chung Cheng University, Chiayi 62102, Taiwan; lanbarla0708@gmail.com

**Keywords:** ionic liquid, nucleophilic aromatic substitution, quartz crystal microbalance, amine gas detection, alcohol gas analysis

## Abstract

We demonstrated in this work the use of affinity ionic liquids, **AIL 1** and **AIL 2**, for chemoselective detection of amine and alcohol gases on a quartz crystal microbalance (QCM). These detections of gaseous amines and alcohols were achieved by nucleophilic aromatic substitution reactions with the electrophilic 1,3,5-triazine-based **AIL 1** thin-coated on quartz chips. Starting with inexpensive reagents, bicyclic imidazolium ionic liquids **AIL 1** and **AIL 2** were readily synthesized in six and four synthetic steps with high isolated yields: 51% and 63%, respectively. The QCM platform developed in this work is readily applicable and highly sensitive to low molecular weight amine gases: for isobutylamine gas (a bacterial volatile) at 10 Hz decrease in resonance frequency (i.e., Δ*F* = −10 Hz), the detectability using **AIL 1** was 6.3 ppb. Our preliminary investigation on detection of the much less nucleophilic alcohol gas by **AIL 1** was also promising. To our knowledge, no example to date of reports based on nucleophilic aromatic substitution reactions demonstrating sensitive gas detection in these triazine ionic liquids on a QCM has been reported.

## 1. Introduction

This work describes an ionic liquid system specifically developed for the detection of amine and alcohol gases by quartz crystal microbalance (QCM) in real-time and at ambient temperature. This label-free and chemoselective QCM detection of amine and alcohol gases was achieved by nucleophilic aromatic substitution (S_N_Ar) reactions with affinity ionic liquid **AIL 1** thin-coated on quartz chips (Figure 1). The structure of **AIL 1** is engineered with a super electrophilic 2,4-dichloro-1,3,5-triazine group responsible for capturing amine and alcohol vapor. Upon S_N_Ar reaction of **AIL 1** with a nucleophilic amine or alcohol gas, a resonance frequency drop is readily detected by a mass-sensitive QCM.

QCM has been in constant focus of research on chemical analysis in both solutions and gas phases over the years [1,2,3]. This is due primarily to its low cost and real-time analysis of adsorption or recognition processes onto the surface of affinity materials such as ionic liquids deposited on electrodes when being operated at room temperature.

Ionic liquids are low-melting organic molten salts composed entirely of cations and anions [4,5]. Organic cations often are nitrogen- or phosphorus-containing compounds that are weakly associated with the anions, often resulting in liquidous salts at ambient temperature. Typically, one of the ions has a delocalized charge such that the formation of the stable crystal lattice is prevented and electrostatic interactions enfold both ions together. These liquids are green solvents because of their ability to dissolve a wide range of chemicals and their very low volatilities and thus negligible vapor pressures.

Cyanuric chloride (2,4,6-trichloro-1,3,5-triazine) and its derivatives are well known for their wide use in pesticide industry, organic synthesis [6,7], and peptide chemistry [8]. Although this class of compounds has been well recognized for their usefulness in chemical applications, its potential in chemoselective gas analysis remains unexplored. We envisioned that nonvolatile ionic liquids embedded with this dichlorotriazine group would be excellent for use in analysis of both amine and alcohol gases. Herein we are reporting our preliminary results on the study of QCM detections of both alcohol and amine gases with a 1,3,5-triazine-conjugated **AIL 1** by S_N_Ar reactions.

The S_N_Ar reaction typically involves a nucleophilic addition to an arene in polar aprotic solvents followed by the elimination of a leaving group. For a smooth S_N_Ar reaction with stabilization of its anionic reaction intermediate (the Meisenheimer complex), it usually requires the presence of at least one electron-withdrawing substituent on the aromatic ring [9]. It is also reported in literature that S_N_Ar reactions proceed with faster reaction rates in ionic liquids than those performed in molecular solvents [10,11,12,13]. These rate accelerations were primarily attributed to both the enhanced nucleophilicity of the nucleophiles such as amines in ionic liquids and the favorable π^+^–π interactions between cations of ionic liquids and aromatic rings present on the substrates, rendering a more positive charge on the reacting *ipso* carbon [14]. Moreover, with comparison to the electrostatically neutral reactants, ionic liquids stabilize the charged transition states and also the intermediate Meisenheimer complexes formed [15]. Based specifically on S_N_Ar reactions, this work reports the synthesis of affinity ionic liquids **AIL** 1–2 and demonstrates their usefulness in amine and alcohol gases detection on QCM.

## 2. Results and Discussion

Scheme 1A illustrates the synthesis of **AIL 1** of which the affinity element could be readily assembled from the commercially inexpensive cyanuric chloride (**3**), with a freshly prepared amine ionic liquid at ambient temperature. We commenced our synthesis of affinity ionic liquid **AIL 1** from an inexpensive 2-(2-aminoethoxy)ethanol (**1**), which underwent a series of reactions (i.e., amine protection, alcohol mesylation, nucleophilic substitution with a bicyclic 6,7-dihydro-5*H*-pyrrolo[1,2-*a*]imidazole (**2**) previously developed in our laboratory [16], amine deprotection, and, lastly, anion methathesis with lithium bis(trifluoromethane)sulfonimide (LiNTf_2_) in water) to afford first an amine-functionalized ionic liquid. This ionic liquid then reacted with cyanuric chloride (**3**) through a S_N_Ar reaction, as the last step in synthesis, to finally achieve the desired **AIL 1**. Using a different synthetic approach shown in Scheme 1B, the control ionic liquid **AIL 2** was assembled first from the S_N_Ar reaction of the amine **1** with commercial 2-chloro-4,6-dimethoxy-1,3,5-triazine (**4**) under heated conditions followed by alcohol mesylation, nucleophilic substitution with **2**, and lastly an anion metathesis to give **AIL 2**. The syntheses were straightforward and, in our hands, the overall isolated yields for these 6- and 4-step syntheses of **AIL 1** and **AIL 2** were high: 51% and 63%, respectively. Both AILs are viscous pale yellow liquid once obtained and become off-white solid if stored at room temperature for long hours (>1 day), which shows typical properties of supercooled fluids (Appendix A) [17,18]. H-1 and C-13 NMR, and high-resolution mass spectrometry (HRMS) spectra of **AIL 1** and **2** are summarized in the Appendix A.

Analysis and detection of volatile organic compounds (VOCs) are greatly important for a myriad of applications in health, safety, and environment. Moreover, continuous analysis of volatile biogenic amines is important for food spoilage detection (<300 ppb) [19,20,21]. To selectively detect gaseous amines and alcohols, in this work, we incorporated 2,4-dichloro-1,3,5-triazine group in **AIL 1** for reasons that it is a super electrophile and has also been studied extensively in chemistry literature and showed its potential value in target gases detection on QCM.

When injected, amine gas is diffused into a flow QCM device, then dissolved and reacts chemoselectively with **AIL 1**, the frequency shift detected by a QCM is proportional to the mass change of ionic liquid film coated onto a quartz resonator. In principle, the faster the flow rate of the nitrogen carrier gas is, the lesser time an amine gas can react with **AIL 1** and, accordingly, the lower the detectability shall result. For an empirical observation that the flow rate of 3 mL/min in our QCM measurements readily established stable baselines and experimentally produced reproducible sensorgrams, we decided to maintain this slow flow rate carrying a compromised long reaction time but without sacrificing detection sensitivity. Figure 2 displays the QCM results. It is clearly demonstrated that, using the same concentration (100 ppb) for all gaseous samples tested, **AIL 1** only reacted chemoselectively with amine gas: Δ*F* = −175 Hz for isobutylamine [22]. This reaction-based AIL-on-QCM system worked well and was totally unreactive to common VOCs gases such as water, acetone, methanol, ethyl acetate, THF, and hexane (Δ*F* ~ 0 Hz), and, as a result, the frequency drop in this continuous flow measurement was not at all due to any nonspecific physisorption of gas onto ionic liquid. It is worth mentioning that, in our hand, this hydrophobic AIL-on-QCM platform developed was insensitive to moisture, indicating that any residual water present in samples and in the nitrogen gas stream would not interfere with amine gas analysis. As the control affinity ionic liquid, **AIL 2** is inert to isobutylamine and all other gases tested (Figure 2), clearly showing the chemoselectivity of **AIL 1** toward amine gases. An electrospray ionization–high-resolution mass spectrometry (ESI-HRMS) analysis supported the formation of the product of **AIL 1** reaction with isobutylamine gas on quartz chip: new, correct masses of 380.1950 ([M]^+^, 100%), 382.1926 ([M + 2]^+^, 34%) for [C_17_H_27_ClN_7_O]^+^ ion were experimentally obtained (Appendix A).

In addition to isobutylamine, we also investigated other volatile amines (2-methoxyethylamine, isoamylamine, isopropylamine, ethylmethylamine, ethylamine, propylamine, and dimethylamine) for their gas-phase reactions (100 ppb each) with **AIL 1** and tested its effectiveness as a label-free amine gas sensing material. Figure 3 showed that **AIL 1** reacted well with both primary and secondary amines. Among all amines tested, the most reactive dimethylamine (a breath biomarker for renal disease [19]) gas produced the largest Δ*F* value (−295 Hz) with **AIL 1**. Moreover, the ESI-HRMS analysis further verified the product formation of **AIL 1** with a representative propylamine gas on quartz chip: new, correct masses of 366.1814 ([M]^+^, 100%), 368.1771 ([M + 2]^+^, 29%) for [C_16_H_25_ClN_7_O]^+^ ion were experimentally obtained (Appendix A).

As the 2,4-dichloro-1,3,5-triazine-conjugated **AIL 1** was successfully developed for sensitive analysis of amine gases, we next turned our attention to a detailed quantitative study of **AIL 1** and **AIL 2** reactions with isobutylamine gas based on S_N_Ar reactions by QCM. Figure 4 provides the results. The **AIL 1** showed an essentially linear QCM frequency response within the range of concentrations tested (0–300 ppb) (blue sensorgrams in Figure 4A). In our hand, **AIL 1** is highly sensitive to amine gas: for isobutylamine gas at 10 Hz decrease in resonance frequency (i.e., Δ*F* = −10 Hz), the detectability using **AIL 1** was 6.3 ppb (10/1.5874) (blue fitting line in Figure 4B). The QCM responses (Δ*F*) of the control **AIL 2** reactions with isobutylamine gas (0–300 ppb) were completely negligible (red sensorgrams in Figure 4A and red fitting line in Figure 4B).

The aforementioned QCM results further prompted us to initiate a preliminary investigation of the detection of alcohol gases by **AIL 1**. Since alcohols are much less reactive than amines, we envisioned that Lewis acids should make **AIL 1** become more electrophilic and facilitate their reactions. Pleasingly, encouraging results were obtained when Sc(OTf)_3_ was used with **AIL 1** in alcohol gas detection. Our preliminary result is given in Figure 5. Albeit **AIL 1** detects low concentrations of ethanol gas (250 and 500 ppb), we found that 10 mol% Sc(OTf)_3_ Lewis acid further promoted the reaction of **AIL 1** with ethanol gas: at 250 ppb, Δ*F* = −52 Hz with Sc(OTf)_3_ and −14 Hz with no Lewis acid; at 500 ppb, Δ*F* = −83 Hz with Sc(OTf)_3_ and −20 Hz with no Lewis acid, respectively. Results in Figure 5 also unambiguously show that **AIL 2** was inert to ethanol gas (Δ*F* = 0 Hz at 250 and 500 ppb), further proving that the frequency drop in the continuous flow QCM measurement was not at all due to the nonspecific dissolution of ethanol gas in **AIL 1**. This preliminary result revealed that **AIL 1** is chemoselective and **AIL 2** is inert to ethanol gas (Figure 5). The mass analysis supported the formation of S_N_Ar product of **AIL 1** reaction with ethanol: new, correct masses of 353.1479 ([M]^+^, 100%), 355.1454 ([M + 2]^+^, 33%) for [C_15_H_22_ClN_6_O_2_]^+^ ion were experimentally obtained (Appendix A).

## 3. Materials and Methods

### 3.1. General Information

The ^1^H-NMR and ^13^C-NMR spectra were recorded at 400 MHz and 100 MHz, respectively, on a Bruker AVANCEIII HD 400 NMR spectrometer (Bruker BioSpin GmbH, Rheinstetten, Germany) in deuterated solvents (CDCl_3_, DMSO-*d*_6_ or CD_3_CN) using tetramethylsilane (TMS) as the internal standard. The chemical shift (*δ*) for ^1^H and ^13^C are given in ppm relative to the residual signal of the solvent. Coupling constants are given in Hz. The following abbreviations are used to indicate the multiplicity: s, singlet; d, doublet; t, triplet; q, quartet; quin, quintet; m, multiplet; dd, doublet of doublets; td, triplet of doublets; dt, doublet of triplets; ddd, doublet of doublet of doublets; bs, broad signal. The reactions were monitored using TLC (thin-layer chromatography) silica gel 60 F254 (Merck KGaA, Darmstadt, Germany). Evaporation of solvents was performed under reduced pressure. Melting points were measured and recorded by the OptiMelt MPA-100 apparatus (Standford Research Systems, Sunnyvale, CA, USA) and uncorrected.

In addition to cyanuric chloride, all starting materials, reagents, and solvents were purchased from commercial sources and used as such without further purification. Unless otherwise stated, all reactions were carried out under an inert atmosphere using anhydrous solvents.

### 3.2. Synthesis of Affinity Ionic Liquid ***AIL 1***

#### 3.2.1. Synthesis of *tert*-Butyl (2-(2-Hydroxyethoxy)ethyl)carbamate

To a round-bottomed flask containing dichloromethane (15 mL), 2-(2-aminoethoxy)ethanol **1** (1.580 g, 15.03 mmol) was added. The solution was kept in ice bath (0 °C). A dichloromethane solution (9 mL) containing di-*tert*-butyl dicarbonate (3.726 g, 17.07 mmol) was added dropwise (25 min) to the round-bottomed flask. The progress of this amine protection reaction could be conveniently monitored using TLC. The reaction was carried out at room temperature for 6.5 h.

After the reaction, the mixture was washed first with 10 wt% citric acid (10 mL), then 10 wt% NaHCO_3_ (10 mL × 2), and finally with water (10 mL). The organic layer collected was dried over anhydrous Na_2_SO_4_, filtered and concentrated under reduced pressure. The residue was purified by column chromatography (ethyl acetate/hexane = 1:3, *v*/*v*) to afford *tert*-butyl (2-(2-hydroxyethoxy)ethyl)carbamate (2.646 g, 86% yield) as colorless liquid.

#### 3.2.2. Synthesis of *tert*-Butyl (2-(2-(Methylsulfonyl)oxyethoxy)ethyl)carbamate

A solution of methanesulfonyl chloride (1.25 mL, 16.12 mmol) in dichloromethane (14 mL) was slowly added to a ice chilled solution of *tert*-butyl (2-(2-hydroxyethoxy)ethyl)carbamate (2.643 g, 12.87 mmol) and triethylamine (3.6 mL, 25.75 mmol) in dichloromethane (30.0 mL) in a round-bottomed flask. The resulting solution was allowed to react at 0 °C for 30 min and then brought to ambient temperature for another 2 h.

The reaction solution was first diluted with dichloromethane (11 mL), then sequentially extracted with 10 wt% citric acid (20 mL × 3), 10 wt% NaHCO_3_ (20 mL × 3), finally dried over Na_2_SO_4_. After filtration, the solvent was removed under reduced pressure and concentrated in vacuo to afford high purity product, *tert*-butyl (2-(2-(methylsulfonyl)oxyethoxy)ethyl)carbamate, as a yellow liquid (3.646 g, quantitative yield).

#### 3.2.3. Synthesis of Boc-Protected Amine Ionic Liquid

*Tert*-butyl (2-(2-(methylsulfonyl)oxyethoxy)ethyl)carbamate (715 mg, 2.53 mmol) was added to a round-bottomed flask containing 6,7-dihydro-5*H*-pyrrolo[1,2-*a*]imidazole **2** (268 mg, 2.48 mmol) previously developed in our laboratory [16]. The reaction progress was conveniently monitored by TLC, and the alkylation reaction was carried out at 105 °C for 4 h.

The desired product in reaction mixture was purified by silica gel column chromatography, first, using eluents containing ethyl acetate/hexane = 1:1 to 1:0 (*v*/*v*), and then changing mobile phase to methanol to eventually afford the Boc-protected amine ionic liquid (0.903 g, 93% yield) as a colorless liquid.

#### 3.2.4. Deprotection of Boc-Protected Amine Ionic Liquid

To a 20-mL sample bottle containing Boc-protected amine ionic liquid (824 mg, 2.11 mmol) in dichloromethane (9.7 mL), trifluoroacetic acid (0.8 mL, 137.52 mmol) was added for Boc-deprotection. The reaction progress was monitored by TLC and its reaction was carried out for 13.5 h at ambient temperature. The solvent and excess TFA was removed under reduced pressure and concentrated in vacuo for 5 h to afford a yellow waxy product without further purification.

#### 3.2.5. Synthesis of Amine Ionic Liquid

To an ice chilled container containing the yellow waxy compound was first added 10 wt% NaHCO_3_ (2.2 mL), followed by LiNTf_2_ (1.222 g, 4.26 mmol). The solution mixture became two phases within 5 min in which the desired hydrophobic amine ionic liquid was in the bottom phase. The solution was stirred for a total of 14 h at ambient temperature. The bottom ionic liquid was collected, washed with water and dichloromethane, and concentrated in vacuo to finally afford the amine ionic liquid as yellow liquid with quantitative isolated yield (1.00 g). Yellow liquid; ^1^H-NMR (400 MHz, CD_3_CN) δ 2.73 (quin, *J* = 7.5 Hz, ImCH_2_C*H*_2_, 2H), 3.01 (t, *J* = 5.1 Hz, C*H*_2_NH_2_, 2H), 3.10 (t, *J* = 7.8 Hz, NC(C*H*_2_)=N, 2H), 3.56 (t, *J* = 5.1 Hz, OC*H*_2_CH_2_NH_2_, 2H), 3.75 (t, *J* = 5.0 Hz, ImCH_2_C*H*_2_O, 2H), 3.85–4.29 (bs, NH_2_, 2H), 4.17 (t, *J* = 5.0 Hz, ImC*H*_2_CH_2_O, 2H), 4.20 (t, *J* = 7.2 Hz, ImC*H*_2_, 2H), 7.28 (s, Im-H, 1H), 7.31 (s, Im-H, 1H); ^13^C-NMR (100 MHz, DMSO-*d*_6_) δ 22.74, 25.54, 41.13, 47.87, 48.23, 68.12, 72.86, 117.76, 119.45 (q, *J*_CF_ = 319 Hz), 126.03, 152.79.

#### 3.2.6. Synthesis of Affinity Ionic Liquid **AIL 1**

To a round-bottomed flask containing cyanuric chloride **3** (178 mg, 0.97 mmol) and NaHCO_3_ (69 mg, 0.82 mmol) in acetonitrile (3.0 mL) at 0 °C, a acetonitrile solution (5 mL) containing the amine ionic liquid (256 mg, 0.54 mmol) was added dropwise for a period of 30 min. The nucleophilic aromatic substitution reaction was carried out at ambient temperature for 4 h, determined by TLC.

Salts formed in the reaction mixture were filtered off. The solvent was removed under reduced pressure and concentrated in vacuo for 30 min. The crude product was mixed with dichloromethane (30 mL) and washed with 10 wt% citric acid (15 mL × 3). The organic layer collected was dried over anhydrous Na_2_SO_4_, filtered and concentrated in vacuo for 3 h. The desired product was thoroughly washed with n-hexane, followed by chloroform to remove excessive cyanuric chloride **3** and its hydrolysis impurities to finally afford the pure **AIL 1** (219 mg, 65% yield) as ivory white solid.

Ivory white solid, mp 74 °C; ^1^H-NMR (400 MHz, CD_3_CN) δ 2.71 (quin, *J* = 7.5 Hz, ImCH_2_C*H*_2_, 2H), 3.08 (t, *J* = 7.6 Hz, NC(C*H*_2_)=N, 2H), 3.50–3.61 (m, OC*H*_2_C*H*_2_NH, 4H), 3.73 (t, *J* = 4.8 Hz, ImCH_2_C*H*_2_O, 2H), 4.15 (t, *J* = 4.8 Hz, ImC*H*_2_CH_2_O, 2H), 4.19 (t, *J* = 7.4 Hz, ImC*H*_2_, 2H), 6.87–7.03 (bs, NH, 1H), 7.25 (d, *J* = 2.2 Hz, Im-H, 1H), 7.27 (d, *J* = 2.2 Hz, Im-H, 1H); ^13^C-NMR (100 MHz, CD_3_CN) δ 24.12, 26.64, 41.82, 49.28, 49.92, 69.45, 69.53, 118.66, 121.02 (q, *J*_CF_ = 319 Hz), 127.05, 154.41, 167.32, 170.54, 171.33; ESI-HRMS *m*/*z* [M]^+^ calculated for C_13_H_17_Cl_2_N_6_O 343.0841, found 343.0844 ([M]^+^, 100%), 345.0804 ([M + 2]^+^, 66%), 347.0810 ([M + 4]^+^, 12%).

### 3.3. Synthesis of Affinity Ionic Liquid ***AIL 2***

#### 3.3.1. Synthesis of 2-(2-((4,6-Dimethoxy-1,3,5-triazin-2-yl)amino)ethoxy)ethan-1-ol

To a round-bottomed flask containing 2-chloro-4,6-dimethoxy-1,3,5-triazine **4** (435 mg, 3.28 mmol) in acetonitrile (10 mL) at 0 °C was added dropwise a solution of 2-(2-aminoethoxy)ethanol **1** (240 mg, 2.28 mmol) in acetonitrile (5 mL) for a period of 1 h. The reaction solution was then mixed with NaHCO_3_ (210 mg, 2.51 mmol). The progress of this nucleophilic aromatic substitution reaction could be readily monitored by TLC and was proceeded at 83 °C for a total of 16.5 h.

After the reaction, salts formed were filtered and the solution was concentrated in vacuo for 30 min. The crude product was then mixed with dichloromethane (30 mL), and the resulting solution was extracted with 10 wt% sodium bicarbonate (15 mL × 3). The organic layer collected was dried over anhydrous Na_2_SO_4_, filtered and concentrated under reduced pressure and in vacuo for another 3 h. Purification was done by silica gel column chromatography (ethyl acetate/hexane = 1:3 to 1:0, *v*/*v*) and the desired compound, 2-(2-((4,6-dimethoxy-1,3,5-triazin-2-yl)amino)ethoxy)ethan-1-ol was isolated as a white solid (438 mg, 80% yield).

#### 3.3.2. Synthesis of 2-(2-((4,6-Dimethoxy-1,3,5-triazin-2-yl)amino)ethoxy)ethyl methanesulfonate

To a round-bottomed flask containing 2-(2-((4,6-dimethoxy-1,3,5-triazin-2-yl)amino)ethoxy) ethan-1-ol (438 mg, 1.79 mmol) and triethylamine (0.5 mL, 3.59 mmol) in dichloromethane (5 mL) at 0 °C, a solution of methanesulfonyl chloride (310 mg, 2.71 mmol) in dichloromethane (2.4 mL) was added slowly. The resulting mixture was allowed to react in ice bath and then at ambient temperature for 2.5 h.

The resulting reaction solution was further diluted with dichloromethane (14 mL), then sequentially extracted with 10 wt% citric acid (7 mL × 3) and 10 wt% NaHCO_3_ (7 mL × 3). The organic layer collected was dried over anhydrous Na_2_SO_4_, filtered, concentrated under reduced pressure and in vacuo to afford a light yellow liquid product (563 mg, 97% yield).

#### 3.3.3. Synthesis of Triazine Ionic Liquid

2-(2-((4,6-dimethoxy-1,3,5-triazin-2-yl)amino)ethoxy)ethyl methanesulfonate (117 mg, 0.36 mmol) was added to a round-bottomed flask containing 6,7-dihydro-5H-pyrrolo[1,2-*a*]imidazole **2** (35 mg, 0.33 mmol) previously developed in our laboratory (*Tetrahedron*
**2007**, *63*, 1644–1653). The reaction progress was conveniently monitored by TLC and this alkylation reaction was carried out at 72 °C for 25 h.

The desired product in reaction mixture was purified by silica gel column chromatography, first, using eluents containing ethyl acetate/hexane = 1:1 to 1:0 (*v*/*v*), and then changing mobile phase to methanol to eventually afford the triazine ionic liquid (116 mg, 83% yield) as colorless liquid.

#### 3.3.4. Synthesis of **AIL 2**

To a ice chilled solution of the triazine ionic liquid (112 mg, 0.26 mmol), LiNTf_2_ (90 mg, 0.31 mmol) in water (0.4 mL) and dichloromethane (0.4 mL) was added The ion-exchange reaction was vigorously stirred for 13 h at ambient temperature. The upper aqueous layer was removed, and the bottom dichloromethane solution was extracted with water (1 mL × 4). The organic layer collected was dried over anhydrous Na_2_SO_4_, filtered, and concentrated under reduced pressure to eventually afford the desired **AIL2** (158 mg, 98% yield) as white solid.

White solid, mp 79 °C; ^1^H-NMR (400 MHz, CDCl_3_) δ 2.84 (quin, *J* = 7.5 Hz, ImCH_2_C*H*_2_, 2H), 3.23 (t, *J* = 7.6 Hz, NC(C*H*_2_)=N, 2H), 3.59–3.69 (m, OC*H*_2_C*H*_2_NH, 4H), 3.81 (t, *J* = 4.8 Hz, ImCH_2_C*H*_2_O, 2H), 3.94 (s, O*H*_3_, 3H) , 3.96 (s, O*H*_3_, 3H), 4.24 (t, *J* = 4.8 Hz, ImC*H*_2_CH_2_O, 2H), 4.31 (t, *J* = 7.4 Hz, ImC*H*_2_, 2H), 5.64–5.74 (bs, NH, 1H), 7.20 (d, *J* = 2.20 Hz, Im-H, 1H), 7.31 (d, *J* = 2.0 Hz, Im-H, 1H); ^13^C-NMR (100 MHz, CDCl_3_) δ 23.32, 25.88, 40.64, 48.48, 49.59, 54.84, 54.88, 68.75, 69.76, 117.80, 119.99 (q, *J*_CF_ = 319 Hz), 126.48, 153.03, 168.49, 172.28, 172.75; ESI-HRMS m/z [M]^+^ calculated for C_15_H_23_N_6_O_3_ 335.1832, found 335.1826.

### 3.4. QCM Measurements

A flow PSS (Piezo Sensor System) QCM system (9 MHz) available from the ANT Technology Co. (Taipei, Taiwan) was operated at room temperature and nitrogen was used as carrier gas. The 9-MHz AT-cut quartz chips deposited with gold electrodes (area 11 mm^2^) on both sides were also available from ANT Technology. Before use, the gold electrodes on chips were cleaned sequentially with 5 N NaOH (30 min), H_2_O (5 min), 1 N HCl (5 min), and H_2_O (5 min). This sequence of chip cleaning was repeated two times, followed by washing with MeOH (30 min), H_2_O (10 min), and H_2_O (10 min) to remove any organic absorbents, and finally rinsed with water thoroughly and dried under nitrogen.

Ionic liquid solutions were prepared by dissolving individual AIL (10 μmol) in acetonitrile (HPLC grade, 300 μL). The freshly prepared solutions (1 μL) were carefully pipetted onto the cleaned bare gold electrode situated at the center of quartz chips. The ionic liquid-coated chips were placed in an oven (120 °C) for 1 min to remove residual solvent. The quartz sensor chips were then mounted in the gas flow chamber (100 cm^3^) and nitrogen was used as carrier gas at flow rate of 3 mL/min. Until a stable baseline was obtained, target gas samples were injected into the chamber. The resonance frequency drops versus time curves were measured and recorded. Typically, the used quartz chips could be readily regenerated by a simple wash with acetonitrile or methanol solvent.

Target sample vapors were obtained by gasifying the chemicals in the sealed glass container (1.25 L). Using the PSS QCM apparatus for QCM measurements, a rapid initial nonspecific frequency decrease (approximately 0.1 to 0.4 Hz) was typically detected within 5 s after sample injection, which was totally insignificant with reference to reaction-based frequency drops owing to nucleophilic aromatic substitution reactions of AILs with target gases.

## 4. Conclusions

We demonstrated in this work the use of affinity ionic liquids, **AIL 1** and **AIL 2**, for chemoselective detection of amine [23] and alcohol [24] gases on a quartz crystal microbalance. These detections of gaseous amines and alcohols were achieved by S_N_Ar reactions with the super electrophilic 1,3,5-triazine-based **AIL 1** thin-coated on quartz chips. Our work holds three important conclusions: (i) affinity ionic liquid **AIL 1** developed in this work is highly sensitive to the chemoselective detection of amine gases, (ii) our QCM analysis of S_N_Ar reactions by **AIL 1** is straightforward and label-free, and (iii) our ionic liquid platform is chemoselective, readily applicable to analysis of small molecular weight amine and alcohol gases. To the best of our knowledge, we know of no examples to date of reports based on S_N_Ar reactions demonstrating sensitive amine as well as alcohol gas detection with 1,3,5-triazine ionic liquids on a QCM.

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
