# Peer review of "Reaction-Based Amine and Alcohol Gases Detection with Triazine Ionic Liquid Materials"

_molecules, 2019, doi:10.3390/molecules25010104_

Round 1
Reviewer 1 Report
The authors demonstrated a novel approach to detect amine and alcohol gases via affinity ionic liquid coated quartz. The manuscript was well written and experiments are carefully designed. The results are interesting and are important for sensor applications. There are only few points to be addressed before accepting this manuscript:
In the manuscript, Line 43, it was mentioned "cations are weakly associated to anions." Ionic liquids, in comparison with aqueous electrolytes (in which the charge of ions are screened by solvent molecules) have strong cation-anion interaction. Please be more specific as to what you are referencing when you mention "weakly associated." Please mention in the introduction, which level of sensitivity (in ppb) is required for the sensor in the applications mentioned in the Line 90-100. The linear relationship between amine concentration and frequency change of AIL1-coated quartz allows easy extraction of precise concentration, which makes AIL1 a very interesting materials. However, for medical applications, the reaction time is crucial. Please comment on the detection time. How to improve it? Will thicker AIL1 deposition increase the sensitivity and/or reaction time? Or it will rather increase the noise level and eventually lose the sensitivity? Please comment.Author Response
December 15, 2019
MS: “Reaction-Based Amine and Alcohol Gases Detection with Triazine Ionic Liquid Materials”
Authors: Hsin-Yi Li and Yen-Ho Chu*
Figures: 5 (+ S4); Scheme: 1; Supporting Information: 1
Dear Editor,
Enclosed please find our revised manuscript entitled “Reaction-Based Amine and Alcohol Gases Detection with Triazine Ionic Liquid Materials” that we would like to submit for publication in the Molecules.
First, we would like to thank all four reviewers for their reviews of our manuscript and their constructive comments. In this revised manuscript, we have incorporated all points addressed by all reviewers. Specifically, we have addressed the following points raised by the reviewer 1:
“The authors demonstrated a novel approach to detect amine and alcohol gases via affinity ionic liquid coated quartz. The manuscript was well written and experiments are carefully designed. The results are interesting and are important for sensor applications. There are only few points to be addressed before accepting this manuscript:
In the manuscript, line 43, it was mentioned “cations are weakly associated to anions.” Ionic liquids, in comparison with aqueous electrolytes (in which the charge of ions are screened by solvent molecules) have strong cation-anion interaction. Please be more specific as to what you are referencing when you mention “weakly associated”.”
We thank the reviewer for constructive comment. As the reviewer pointed out and we totally agreed, we have accordingly addressed the point by rephrasing the original sentence to complete its meaning (line 43, page 2), labeled in red, in the revised manuscript.
“Please mention in the introduction, which level of sensitivity (in ppb) is required for the sensor in the applications mentioned in the line 90-100.”
As the reviewer pointed out and we agreed, we have accordingly addressed the point by providing levels of sensitivity for breath acetone, ammonia, and volatile organic amines (lines 98 and 101, page 3), labeled in red, in the revised manuscript.
“The linear relationship between amine concentration and frequency change of AIL 1-coated quartz allows easy extraction of precise concentration, which makes AIL 1 a very interesting materials. However, for medical applications, the reaction time is crucial. Please comment on the detection time. How to improve it? Will thicker AIL 1 deposition increase the sensitivity and/or reaction time? Or, it will rather increase the noise level and eventually lose sensitivity? Please comment.”
We thank the reviewer for the constructive comment. As the reviewer pointed out and we agreed, we have accordingly addressed the point by (i) rephrasing our sentence and (ii) adding two new sentences (lines 105-112, page 3), labeled in red, in the revised manuscript.
We have also addressed the following points raised by the reviewer 2:
“This work is mainly focused on i) the synthesis of triazine based ionic liquids and ii) the characterization of the reaction products resulting from the interaction of ionic liquids with amine and alcohols. In my opinion, the QCM platform reveals as valuable characterization tool for monitoring such interactions more than a gas sensor for practical applications. Thus why I consider appropriate to re-orientate the manuscript to a more generic text.
My main concerns are related to the following:
-The high toxicity of some of the chemicals used for the synthesis of triazine based ionic liquids: cyanuric chloride and methyl cyanide. Although the vapor pressure of the resulting ILs is almost negligible, the authors could comment on the potential harzards and the specific measures to consider when handling such reactions. Also, it seems appropriate to comment on how scalable is the IL synthesis (with yields below 60%) for being implemented an industrial level.”
We thank the reviewer for the comment. As the reviewer pointed out and we totally agreed that we hardly claimed in this work as a sensor for amine and alcohol gases, we only stated that our work was a “reaction-based amine and alcohol gases detection with triazine ionic liquid materials” (our manuscript title) and a “chemoselective detection of amine and alcohol gases by a quartz crystal microbalance” (presented in the Conclusion of manuscript). As also the reviewer pointed out and we agreed, in order to successfully synthesize AILs, our AIL synthesis was inevitably involved volatile solvents and hazardous chemicals. Our work presented, at this stage, was purely an academic research.
‘’-A proper characterization of the mechanical response of the resonator, i.e. quality factor and natural frequency, before and after the deposition of the ionic liquid should be provided in the manuscript. The modification of the QCM sensor with the ionic liquid is done by carefully addition with a pipette of the prepared IL solution to get 33 nm/300 nm of sensing coating. More information about the mass and thickness estimation (SEM analysis, AFM, perfilometer) is required. In addition, the viscosity of the IL, and the viscoelastic behaviour of the IL thin film should be discussed in the manuscript and also considered when applying the Sauerbrey equation to calculate the mass variation changes, i.e. IL loading, from the frequency shift.”
We thank the reviewer for the constructive comments. As the reviewer pointed out and we agreed, we have accordingly addressed the reviewer points by (i) including info on masses of AIL 1 and AIL 2 used in QCM measurements in the caption of Figure 2 (line 130, page 4) and (ii) relocating the section of ‘QCM measurements’ originally deposited in the Supporting Information to create a totally new section paragraphs (lines 324-345, pages 9 and 10), labeled in red, in the revised manuscript.
“-The authors underlies the chemoselective detection of gaseous amines and alcohols, but do not carry out detection experiments with multicomponent mixtures to confirm such selective detection.
-Experimental details about the preparation of N2 stream (3 mL/min) containing 100 ppb of the target analyte are missing.”
We thank the reviewer for constructive comment. As the reviewer pointed out and we agreed, we have accordingly addressed the reviewer point by relocating the section of ‘QCM measurements’ originally deposited in the Supporting Information to create a totally new section paragraphs (lines 324-345, pages 9 and 10) with detailed experimental sample preparation and QCM measurements, labeled in red, in the revised manuscript. Also, our work presented in this work, at this stage, was purely an academic research.
“-“Sensitivity” of detection, “Limit of Detection”, “Response Time” and “Recovery Time” parameters have to be properly defined and calculated from the experiments carried out. A zooming of the “y” axis of Figures 2 to 5 will be clearly beneficial to assess on the noise level. In addition, specific comments on the “reusability” of the sensors, i.e. reaction irresversibility, should be given in the text.
-Please avoid auto-cite in the introduction when describing the QCMs as gas sensors.”
We thank the reviewer for comment. In this work, we only used the term “sensitivity of detection” for the concentration of an analyte such as amine that produced a 10 Hz-decrease in resonance frequency (i.e., ΔF = -10 Hz) (lines 149-151, page 4). Also, the used quartz chips could readily be regenerated by washing with solvents (acetonitrile or methanol) and coating again the AILs (lines 339-340, page 9), labeled in red, in the revised manuscript.
We have also addressed the following points raised by the reviewer 3:
“The manuscript has been well organized, only several minor points need to be revised:
Title: Is correct “Gase detection”?
Abstract: Line 10. Please include here QCM abbreviation after the complete name.
Line 51: …group would be excellent…
Line 113: Include here the complete name for ESI-HRMS. The same for TMS in line 183.”
We thank the reviewer for constructive comments. As the reviewer pointed out and we totally agreed, we have accordingly addressed the reviewer points by (i) correcting the typo at manuscript title (line 2, page 1), (ii) including ‘QCM’ in Abstract (line 10, page 10), (iii) deleting ‘…be…’ and ‘…candidates…’ (line 52, page 2), and (iv) providing the full name for ESI-HRMS (line 123, page 4) and TMS (line 194, page 6), labeled in red, in the revised manuscript.
“Section 3.2. The abbreviation used in Scheme 1 would be very valuable here to easily follow the synthesis procedures. For example, Boc2O, MsCl, TFA …”
References: In my opinion two references should be included: Toniolo et al., Anal. Chem. 85, 2013, 7241-7247. Xu et al., Sensors and Actuators B 134, 2008, 258-265.”
We thank the reviewer for constructive comments. As the reviewer pointed out and we totally agreed, we have accordingly addressed the reviewer points by (i) providing the abbreviated chemicals their full names in the caption of Scheme 1 (lines 91-92, page 3), labeled in red, in the revised manuscript. Also, we agree to add the two references suggested by the reviewer (ref# 26 and 27, page 11), labeled in red, in the revised manuscript.
We have also addressed the following points raised by the reviewer 4:
“This manuscript reported the synthesis of two new ionic liquids and the application of them as chemoselective vapor sensors. The main concern is the sensing responses rely on the nucleophilic aromatic substitution, which gives selective sensing performances. However, on the other hand, this reaction would resulted in the irreversible sensing as the amines cannot be eliminated by purging with N2 gas. The ability of recovery is a key criterion in sensing technology. Therefore, I think the main claim of this manuscript is not acceptable. It is confusing of the statement of sensing of amine: “for isobutylamine gas (a bacterial volatile) at 10 Hz decrease in resonance frequency”. It is described in Figure 2, the response is -175 Hz. How is the sensitivity of detection of 6.3 ppb calculated? In terms of experimental details: 1) How was vapor concentration in ppb level realized in the experiments? How was it calibrated? 2) how were the mass spectrometry experiments performed/ How were the sample prepared, on QCM substrates or bulk synthesis?”
We thank the reviewer for constructive comments. As this and other reviewers pointed out the inquiry of the experimental details in QCM preparation and measurements and we totally agreed, we have accordingly addressed the reviewer points by relocating the section of ‘QCM measurements’ originally deposited in the Supporting Information to create and place a totally new section paragraphs (lines 324-345, pages 9 and 10), labeled in red, in the revised manuscript. Also, as clearly stated in the text (lines 149-151, page 4) as well as the caption of Figure 4 (line 168, page 6), the value of 6.3 ppb was obtained from 10 Hz / 1.5874 (= 6.3 ppb) in the revised manuscript.
We have revised our manuscript and addressed all points raised by all four reviewers. We hope that our manuscript is now in suitable form for publication.
Best wishes.
Very truly yours,
Yen-Ho Chu
Distinguished Professor of Chemistry and Biochemistry
National Chung Cheng University
Chiayi 62102, Taiwan, ROC
Phone: +886-5-272-0411 ext. 66408
Fax: +886-5-272-1040
E-mail: cheyhc@ccu.edu.tw

Reviewer 2 Report
This work is mainly focused on i) the synthesis of triazine based ionic liquids and ii) the characterization of the reaction products resulting from the interaction of ionic liquids with amine and alcohols. In my opinion, the QCM platform reveals as a valuable characterization tool for monitoring such interactions more than a gas sensor for practical applications. Thus why I consider appropriate to re-orientate the manuscript to a more generic context.
My main concerns are related to the following:
-The high toxicity of some of the chemicals used for the synthesis of triazine based ionic liquids: cyanuric chloride and methyl cyanide. Although the vapor pressure of the resulting ILs is almost negligible, the authors could comment on the potential hazards and the specific measures to consider when handling such reactants. Also, it seems appropriate to comment on how scalable is the IL synthesis (with yields below 60%) for being implemented at industrial level.
-A proper characterization of the mechanical response of the resonator, i.e. quality factor and natural frequency, before and after the deposition of the ionic liquid should be provided in the manuscript. The modification of the QCM sensor with the ionic liquid is done by carefully addition with a pipette of the prepared IL solution to get 33 nm / 300 nm of sensing coating. More information about the mass and thickness estimation (SEM analysis, AFM, perfilometer) is required. In addition, the viscosity of the IL, and the viscoelastic behaviour of the IL thin film should be discussed in the manuscript and also considered when applying the Sauerbrey equation to calculate the mass variation changes, i.e. IL loading, from the frequency shift.
-The authors underlies the chemoselective detection of gaseous amines and alcohols, but do not carry out detection experiments with multicomponent mixtures to confirm such selective detection.
-Experimental details about the preparation of the N2 stream (3mL/min) containing 100 ppb of the target analyte are missing.
-"Sensitivity" of detection, "Limit of Detection", "Response Time" and "Recovery Time" parameters have to be properly defined and calculated from the experiments carried out. A zooming of the "y" axis of Figures 2 to 5 will be clearly beneficial to asses on the noise level. In addition, specific comments on the "reusability" of the sensors, i.e. reaction irreversibility, should be given in the text.
-Please avoid auto-cite in the introduction when describing the QCMs as gas sensors.
Author Response
December 15, 2019
MS: “Reaction-Based Amine and Alcohol Gases Detection with Triazine Ionic Liquid Materials”
Authors: Hsin-Yi Li and Yen-Ho Chu*
Figures: 5 (+ S4); Scheme: 1; Supporting Information: 1
Dear Editor,
Enclosed please find our revised manuscript entitled “Reaction-Based Amine and Alcohol Gases Detection with Triazine Ionic Liquid Materials” that we would like to submit for publication in the Molecules.
First, we would like to thank all four reviewers for their reviews of our manuscript and their constructive comments. In this revised manuscript, we have incorporated all points addressed by all reviewers. Specifically, we have addressed the following points raised by the reviewer 1:
“The authors demonstrated a novel approach to detect amine and alcohol gases via affinity ionic liquid coated quartz. The manuscript was well written and experiments are carefully designed. The results are interesting and are important for sensor applications. There are only few points to be addressed before accepting this manuscript:
In the manuscript, line 43, it was mentioned “cations are weakly associated to anions.” Ionic liquids, in comparison with aqueous electrolytes (in which the charge of ions are screened by solvent molecules) have strong cation-anion interaction. Please be more specific as to what you are referencing when you mention “weakly associated”.”
We thank the reviewer for constructive comment. As the reviewer pointed out and we totally agreed, we have accordingly addressed the point by rephrasing the original sentence to complete its meaning (line 43, page 2), labeled in red, in the revised manuscript.
“Please mention in the introduction, which level of sensitivity (in ppb) is required for the sensor in the applications mentioned in the line 90-100.”
As the reviewer pointed out and we agreed, we have accordingly addressed the point by providing levels of sensitivity for breath acetone, ammonia, and volatile organic amines (lines 98 and 101, page 3), labeled in red, in the revised manuscript.
“The linear relationship between amine concentration and frequency change of AIL 1-coated quartz allows easy extraction of precise concentration, which makes AIL 1 a very interesting materials. However, for medical applications, the reaction time is crucial. Please comment on the detection time. How to improve it? Will thicker AIL 1 deposition increase the sensitivity and/or reaction time? Or, it will rather increase the noise level and eventually lose sensitivity? Please comment.”
We thank the reviewer for the constructive comment. As the reviewer pointed out and we agreed, we have accordingly addressed the point by (i) rephrasing our sentence and (ii) adding two new sentences (lines 105-112, page 3), labeled in red, in the revised manuscript.
We have also addressed the following points raised by the reviewer 2:
“This work is mainly focused on i) the synthesis of triazine based ionic liquids and ii) the characterization of the reaction products resulting from the interaction of ionic liquids with amine and alcohols. In my opinion, the QCM platform reveals as valuable characterization tool for monitoring such interactions more than a gas sensor for practical applications. Thus why I consider appropriate to re-orientate the manuscript to a more generic text.
My main concerns are related to the following:
-The high toxicity of some of the chemicals used for the synthesis of triazine based ionic liquids: cyanuric chloride and methyl cyanide. Although the vapor pressure of the resulting ILs is almost negligible, the authors could comment on the potential harzards and the specific measures to consider when handling such reactions. Also, it seems appropriate to comment on how scalable is the IL synthesis (with yields below 60%) for being implemented an industrial level.”
We thank the reviewer for the comment. As the reviewer pointed out and we totally agreed that we hardly claimed in this work as a sensor for amine and alcohol gases, we only stated that our work was a “reaction-based amine and alcohol gases detection with triazine ionic liquid materials” (our manuscript title) and a “chemoselective detection of amine and alcohol gases by a quartz crystal microbalance” (presented in the Conclusion of manuscript). As also the reviewer pointed out and we agreed, in order to successfully synthesize AILs, our AIL synthesis was inevitably involved volatile solvents and hazardous chemicals. Our work presented, at this stage, was purely an academic research.
‘’-A proper characterization of the mechanical response of the resonator, i.e. quality factor and natural frequency, before and after the deposition of the ionic liquid should be provided in the manuscript. The modification of the QCM sensor with the ionic liquid is done by carefully addition with a pipette of the prepared IL solution to get 33 nm/300 nm of sensing coating. More information about the mass and thickness estimation (SEM analysis, AFM, perfilometer) is required. In addition, the viscosity of the IL, and the viscoelastic behaviour of the IL thin film should be discussed in the manuscript and also considered when applying the Sauerbrey equation to calculate the mass variation changes, i.e. IL loading, from the frequency shift.”
We thank the reviewer for the constructive comments. As the reviewer pointed out and we agreed, we have accordingly addressed the reviewer points by (i) including info on masses of AIL 1 and AIL 2 used in QCM measurements in the caption of Figure 2 (line 130, page 4) and (ii) relocating the section of ‘QCM measurements’ originally deposited in the Supporting Information to create a totally new section paragraphs (lines 324-345, pages 9 and 10), labeled in red, in the revised manuscript.
“-The authors underlies the chemoselective detection of gaseous amines and alcohols, but do not carry out detection experiments with multicomponent mixtures to confirm such selective detection.
-Experimental details about the preparation of N2 stream (3 mL/min) containing 100 ppb of the target analyte are missing.”
We thank the reviewer for constructive comment. As the reviewer pointed out and we agreed, we have accordingly addressed the reviewer point by relocating the section of ‘QCM measurements’ originally deposited in the Supporting Information to create a totally new section paragraphs (lines 324-345, pages 9 and 10) with detailed experimental sample preparation and QCM measurements, labeled in red, in the revised manuscript. Also, our work presented in this work, at this stage, was purely an academic research.
“-“Sensitivity” of detection, “Limit of Detection”, “Response Time” and “Recovery Time” parameters have to be properly defined and calculated from the experiments carried out. A zooming of the “y” axis of Figures 2 to 5 will be clearly beneficial to assess on the noise level. In addition, specific comments on the “reusability” of the sensors, i.e. reaction irresversibility, should be given in the text.
-Please avoid auto-cite in the introduction when describing the QCMs as gas sensors.”
We thank the reviewer for comment. In this work, we only used the term “sensitivity of detection” for the concentration of an analyte such as amine that produced a 10 Hz-decrease in resonance frequency (i.e., ΔF = -10 Hz) (lines 149-151, page 4). Also, the used quartz chips could readily be regenerated by washing with solvents (acetonitrile or methanol) and coating again the AILs (lines 339-340, page 9), labeled in red, in the revised manuscript.
We have also addressed the following points raised by the reviewer 3:
“The manuscript has been well organized, only several minor points need to be revised:
Title: Is correct “Gase detection”?
Abstract: Line 10. Please include here QCM abbreviation after the complete name.
Line 51: …group would be excellent…
Line 113: Include here the complete name for ESI-HRMS. The same for TMS in line 183.”
We thank the reviewer for constructive comments. As the reviewer pointed out and we totally agreed, we have accordingly addressed the reviewer points by (i) correcting the typo at manuscript title (line 2, page 1), (ii) including ‘QCM’ in Abstract (line 10, page 10), (iii) deleting ‘…be…’ and ‘…candidates…’ (line 52, page 2), and (iv) providing the full name for ESI-HRMS (line 123, page 4) and TMS (line 194, page 6), labeled in red, in the revised manuscript.
“Section 3.2. The abbreviation used in Scheme 1 would be very valuable here to easily follow the synthesis procedures. For example, Boc2O, MsCl, TFA …”
References: In my opinion two references should be included: Toniolo et al., Anal. Chem. 85, 2013, 7241-7247. Xu et al., Sensors and Actuators B 134, 2008, 258-265.”
We thank the reviewer for constructive comments. As the reviewer pointed out and we totally agreed, we have accordingly addressed the reviewer points by (i) providing the abbreviated chemicals their full names in the caption of Scheme 1 (lines 91-92, page 3), labeled in red, in the revised manuscript. Also, we agree to add the two references suggested by the reviewer (ref# 26 and 27, page 11), labeled in red, in the revised manuscript.
We have also addressed the following points raised by the reviewer 4:
“This manuscript reported the synthesis of two new ionic liquids and the application of them as chemoselective vapor sensors. The main concern is the sensing responses rely on the nucleophilic aromatic substitution, which gives selective sensing performances. However, on the other hand, this reaction would resulted in the irreversible sensing as the amines cannot be eliminated by purging with N2 gas. The ability of recovery is a key criterion in sensing technology. Therefore, I think the main claim of this manuscript is not acceptable. It is confusing of the statement of sensing of amine: “for isobutylamine gas (a bacterial volatile) at 10 Hz decrease in resonance frequency”. It is described in Figure 2, the response is -175 Hz. How is the sensitivity of detection of 6.3 ppb calculated? In terms of experimental details: 1) How was vapor concentration in ppb level realized in the experiments? How was it calibrated? 2) how were the mass spectrometry experiments performed/ How were the sample prepared, on QCM substrates or bulk synthesis?”
We thank the reviewer for constructive comments. As this and other reviewers pointed out the inquiry of the experimental details in QCM preparation and measurements and we totally agreed, we have accordingly addressed the reviewer points by relocating the section of ‘QCM measurements’ originally deposited in the Supporting Information to create and place a totally new section paragraphs (lines 324-345, pages 9 and 10), labeled in red, in the revised manuscript. Also, as clearly stated in the text (lines 149-151, page 4) as well as the caption of Figure 4 (line 168, page 6), the value of 6.3 ppb was obtained from 10 Hz / 1.5874 (= 6.3 ppb) in the revised manuscript.
We have revised our manuscript and addressed all points raised by all four reviewers. We hope that our manuscript is now in suitable form for publication.
Best wishes.
Very truly yours,
Yen-Ho Chu
Distinguished Professor of Chemistry and Biochemistry
National Chung Cheng University
Chiayi 62102, Taiwan, ROC
Phone: +886-5-272-0411 ext. 66408
Fax: +886-5-272-1040
E-mail: cheyhc@ccu.edu.tw

Reviewer 3 Report
The manuscript has been well organized, only several minor points need to be revised:
Title: Is correct "Gase detection"?
Abstract: Line 10. Please include here QMC abbreviation after the complete name.
Line 51: .. group would be excellent...
Line 113: Include here the complete name for ESI-HRMS. The same for TMS in line 183.
Section 3.2. The abbreviation used in Scheme 1 would be very valuable here to easily follow the synthesis procedures. For example, Boc2O, MsCl, TFA...
References: In my opinion two references should be considered to be included:
Toniolo et al., Anal. Chem. 85, 2013, 7241-7247 Xu et al., Sensors and Actuators B 134, 2008, 258-265Author Response
December 15, 2019
MS: “Reaction-Based Amine and Alcohol Gases Detection with Triazine Ionic Liquid Materials”
Authors: Hsin-Yi Li and Yen-Ho Chu*
Figures: 5 (+ S4); Scheme: 1; Supporting Information: 1
Dear Editor,
Enclosed please find our revised manuscript entitled “Reaction-Based Amine and Alcohol Gases Detection with Triazine Ionic Liquid Materials” that we would like to submit for publication in the Molecules.
First, we would like to thank all four reviewers for their reviews of our manuscript and their constructive comments. In this revised manuscript, we have incorporated all points addressed by all reviewers. Specifically, we have addressed the following points raised by the reviewer 1:
“The authors demonstrated a novel approach to detect amine and alcohol gases via affinity ionic liquid coated quartz. The manuscript was well written and experiments are carefully designed. The results are interesting and are important for sensor applications. There are only few points to be addressed before accepting this manuscript:
In the manuscript, line 43, it was mentioned “cations are weakly associated to anions.” Ionic liquids, in comparison with aqueous electrolytes (in which the charge of ions are screened by solvent molecules) have strong cation-anion interaction. Please be more specific as to what you are referencing when you mention “weakly associated”.”
We thank the reviewer for constructive comment. As the reviewer pointed out and we totally agreed, we have accordingly addressed the point by rephrasing the original sentence to complete its meaning (line 43, page 2), labeled in red, in the revised manuscript.
“Please mention in the introduction, which level of sensitivity (in ppb) is required for the sensor in the applications mentioned in the line 90-100.”
As the reviewer pointed out and we agreed, we have accordingly addressed the point by providing levels of sensitivity for breath acetone, ammonia, and volatile organic amines (lines 98 and 101, page 3), labeled in red, in the revised manuscript.
“The linear relationship between amine concentration and frequency change of AIL 1-coated quartz allows easy extraction of precise concentration, which makes AIL 1 a very interesting materials. However, for medical applications, the reaction time is crucial. Please comment on the detection time. How to improve it? Will thicker AIL 1 deposition increase the sensitivity and/or reaction time? Or, it will rather increase the noise level and eventually lose sensitivity? Please comment.”
We thank the reviewer for the constructive comment. As the reviewer pointed out and we agreed, we have accordingly addressed the point by (i) rephrasing our sentence and (ii) adding two new sentences (lines 105-112, page 3), labeled in red, in the revised manuscript.
We have also addressed the following points raised by the reviewer 2:
“This work is mainly focused on i) the synthesis of triazine based ionic liquids and ii) the characterization of the reaction products resulting from the interaction of ionic liquids with amine and alcohols. In my opinion, the QCM platform reveals as valuable characterization tool for monitoring such interactions more than a gas sensor for practical applications. Thus why I consider appropriate to re-orientate the manuscript to a more generic text.
My main concerns are related to the following:
-The high toxicity of some of the chemicals used for the synthesis of triazine based ionic liquids: cyanuric chloride and methyl cyanide. Although the vapor pressure of the resulting ILs is almost negligible, the authors could comment on the potential harzards and the specific measures to consider when handling such reactions. Also, it seems appropriate to comment on how scalable is the IL synthesis (with yields below 60%) for being implemented an industrial level.”
We thank the reviewer for the comment. As the reviewer pointed out and we totally agreed that we hardly claimed in this work as a sensor for amine and alcohol gases, we only stated that our work was a “reaction-based amine and alcohol gases detection with triazine ionic liquid materials” (our manuscript title) and a “chemoselective detection of amine and alcohol gases by a quartz crystal microbalance” (presented in the Conclusion of manuscript). As also the reviewer pointed out and we agreed, in order to successfully synthesize AILs, our AIL synthesis was inevitably involved volatile solvents and hazardous chemicals. Our work presented, at this stage, was purely an academic research.
‘’-A proper characterization of the mechanical response of the resonator, i.e. quality factor and natural frequency, before and after the deposition of the ionic liquid should be provided in the manuscript. The modification of the QCM sensor with the ionic liquid is done by carefully addition with a pipette of the prepared IL solution to get 33 nm/300 nm of sensing coating. More information about the mass and thickness estimation (SEM analysis, AFM, perfilometer) is required. In addition, the viscosity of the IL, and the viscoelastic behaviour of the IL thin film should be discussed in the manuscript and also considered when applying the Sauerbrey equation to calculate the mass variation changes, i.e. IL loading, from the frequency shift.”
We thank the reviewer for the constructive comments. As the reviewer pointed out and we agreed, we have accordingly addressed the reviewer points by (i) including info on masses of AIL 1 and AIL 2 used in QCM measurements in the caption of Figure 2 (line 130, page 4) and (ii) relocating the section of ‘QCM measurements’ originally deposited in the Supporting Information to create a totally new section paragraphs (lines 324-345, pages 9 and 10), labeled in red, in the revised manuscript.
“-The authors underlies the chemoselective detection of gaseous amines and alcohols, but do not carry out detection experiments with multicomponent mixtures to confirm such selective detection.
-Experimental details about the preparation of N2 stream (3 mL/min) containing 100 ppb of the target analyte are missing.”
We thank the reviewer for constructive comment. As the reviewer pointed out and we agreed, we have accordingly addressed the reviewer point by relocating the section of ‘QCM measurements’ originally deposited in the Supporting Information to create a totally new section paragraphs (lines 324-345, pages 9 and 10) with detailed experimental sample preparation and QCM measurements, labeled in red, in the revised manuscript. Also, our work presented in this work, at this stage, was purely an academic research.
“-“Sensitivity” of detection, “Limit of Detection”, “Response Time” and “Recovery Time” parameters have to be properly defined and calculated from the experiments carried out. A zooming of the “y” axis of Figures 2 to 5 will be clearly beneficial to assess on the noise level. In addition, specific comments on the “reusability” of the sensors, i.e. reaction irresversibility, should be given in the text.
-Please avoid auto-cite in the introduction when describing the QCMs as gas sensors.”
We thank the reviewer for comment. In this work, we only used the term “sensitivity of detection” for the concentration of an analyte such as amine that produced a 10 Hz-decrease in resonance frequency (i.e., ΔF = -10 Hz) (lines 149-151, page 4). Also, the used quartz chips could readily be regenerated by washing with solvents (acetonitrile or methanol) and coating again the AILs (lines 339-340, page 9), labeled in red, in the revised manuscript.
We have also addressed the following points raised by the reviewer 3:
“The manuscript has been well organized, only several minor points need to be revised:
Title: Is correct “Gase detection”?
Abstract: Line 10. Please include here QCM abbreviation after the complete name.
Line 51: …group would be excellent…
Line 113: Include here the complete name for ESI-HRMS. The same for TMS in line 183.”
We thank the reviewer for constructive comments. As the reviewer pointed out and we totally agreed, we have accordingly addressed the reviewer points by (i) correcting the typo at manuscript title (line 2, page 1), (ii) including ‘QCM’ in Abstract (line 10, page 10), (iii) deleting ‘…be…’ and ‘…candidates…’ (line 52, page 2), and (iv) providing the full name for ESI-HRMS (line 123, page 4) and TMS (line 194, page 6), labeled in red, in the revised manuscript.
“Section 3.2. The abbreviation used in Scheme 1 would be very valuable here to easily follow the synthesis procedures. For example, Boc2O, MsCl, TFA …”
References: In my opinion two references should be included: Toniolo et al., Anal. Chem. 85, 2013, 7241-7247. Xu et al., Sensors and Actuators B 134, 2008, 258-265.”
We thank the reviewer for constructive comments. As the reviewer pointed out and we totally agreed, we have accordingly addressed the reviewer points by (i) providing the abbreviated chemicals their full names in the caption of Scheme 1 (lines 91-92, page 3), labeled in red, in the revised manuscript. Also, we agree to add the two references suggested by the reviewer (ref# 26 and 27, page 11), labeled in red, in the revised manuscript.
We have also addressed the following points raised by the reviewer 4:
“This manuscript reported the synthesis of two new ionic liquids and the application of them as chemoselective vapor sensors. The main concern is the sensing responses rely on the nucleophilic aromatic substitution, which gives selective sensing performances. However, on the other hand, this reaction would resulted in the irreversible sensing as the amines cannot be eliminated by purging with N2 gas. The ability of recovery is a key criterion in sensing technology. Therefore, I think the main claim of this manuscript is not acceptable. It is confusing of the statement of sensing of amine: “for isobutylamine gas (a bacterial volatile) at 10 Hz decrease in resonance frequency”. It is described in Figure 2, the response is -175 Hz. How is the sensitivity of detection of 6.3 ppb calculated? In terms of experimental details: 1) How was vapor concentration in ppb level realized in the experiments? How was it calibrated? 2) how were the mass spectrometry experiments performed/ How were the sample prepared, on QCM substrates or bulk synthesis?”
We thank the reviewer for constructive comments. As this and other reviewers pointed out the inquiry of the experimental details in QCM preparation and measurements and we totally agreed, we have accordingly addressed the reviewer points by relocating the section of ‘QCM measurements’ originally deposited in the Supporting Information to create and place a totally new section paragraphs (lines 324-345, pages 9 and 10), labeled in red, in the revised manuscript. Also, as clearly stated in the text (lines 149-151, page 4) as well as the caption of Figure 4 (line 168, page 6), the value of 6.3 ppb was obtained from 10 Hz / 1.5874 (= 6.3 ppb) in the revised manuscript.
We have revised our manuscript and addressed all points raised by all four reviewers. We hope that our manuscript is now in suitable form for publication.
Best wishes.
Very truly yours,
Yen-Ho Chu
Distinguished Professor of Chemistry and Biochemistry
National Chung Cheng University
Chiayi 62102, Taiwan, ROC
Phone: +886-5-272-0411 ext. 66408
Fax: +886-5-272-1040
E-mail: cheyhc@ccu.edu.tw

Reviewer 4 Report
This manuscript reported the synthesis of two new ionic liquid and the application of them as chemseletive vapor sensors. The main concern is the sensing responses rely on the nucleophilic aromatic substitution, which gives selective sensing performances. However, on the other hand, this reaction would result in the irreversible sensing as the amines cannot be eliminated by purging with N2 gas. The ability of recovery is a key criterion in sensing technology. Therefore, I think the main claim of this manuscript is not acceptable. It is confusing of the statement of sensing of amine: “for isobutylamine gas (a bacterial volatile) at 10 Hz decrease in resonance frequency”. It is described in Figure 2, the response is -175 Hz. How is the sensitivity of detection of 6.3 ppb calculated? In terms of experimental details: 1) How was vapor concentration in ppb level realized in the experiments? How was it calibrated? 2) How were the mass spectrometry experiments performed? How were the sample prepared, on QCM substrates or bulk synthesis?Author Response
December 15, 2019
MS: “Reaction-Based Amine and Alcohol Gases Detection with Triazine Ionic Liquid Materials”
Authors: Hsin-Yi Li and Yen-Ho Chu*
Figures: 5 (+ S4); Scheme: 1; Supporting Information: 1
Dear Editor,
Enclosed please find our revised manuscript entitled “Reaction-Based Amine and Alcohol Gases Detection with Triazine Ionic Liquid Materials” that we would like to submit for publication in the Molecules.
First, we would like to thank all four reviewers for their reviews of our manuscript and their constructive comments. In this revised manuscript, we have incorporated all points addressed by all reviewers. Specifically, we have addressed the following points raised by the reviewer 1:
“The authors demonstrated a novel approach to detect amine and alcohol gases via affinity ionic liquid coated quartz. The manuscript was well written and experiments are carefully designed. The results are interesting and are important for sensor applications. There are only few points to be addressed before accepting this manuscript:
In the manuscript, line 43, it was mentioned “cations are weakly associated to anions.” Ionic liquids, in comparison with aqueous electrolytes (in which the charge of ions are screened by solvent molecules) have strong cation-anion interaction. Please be more specific as to what you are referencing when you mention “weakly associated”.”
We thank the reviewer for constructive comment. As the reviewer pointed out and we totally agreed, we have accordingly addressed the point by rephrasing the original sentence to complete its meaning (line 43, page 2), labeled in red, in the revised manuscript.
“Please mention in the introduction, which level of sensitivity (in ppb) is required for the sensor in the applications mentioned in the line 90-100.”
As the reviewer pointed out and we agreed, we have accordingly addressed the point by providing levels of sensitivity for breath acetone, ammonia, and volatile organic amines (lines 98 and 101, page 3), labeled in red, in the revised manuscript.
“The linear relationship between amine concentration and frequency change of AIL 1-coated quartz allows easy extraction of precise concentration, which makes AIL 1 a very interesting materials. However, for medical applications, the reaction time is crucial. Please comment on the detection time. How to improve it? Will thicker AIL 1 deposition increase the sensitivity and/or reaction time? Or, it will rather increase the noise level and eventually lose sensitivity? Please comment.”
We thank the reviewer for the constructive comment. As the reviewer pointed out and we agreed, we have accordingly addressed the point by (i) rephrasing our sentence and (ii) adding two new sentences (lines 105-112, page 3), labeled in red, in the revised manuscript.
We have also addressed the following points raised by the reviewer 2:
“This work is mainly focused on i) the synthesis of triazine based ionic liquids and ii) the characterization of the reaction products resulting from the interaction of ionic liquids with amine and alcohols. In my opinion, the QCM platform reveals as valuable characterization tool for monitoring such interactions more than a gas sensor for practical applications. Thus why I consider appropriate to re-orientate the manuscript to a more generic text.
My main concerns are related to the following:
-The high toxicity of some of the chemicals used for the synthesis of triazine based ionic liquids: cyanuric chloride and methyl cyanide. Although the vapor pressure of the resulting ILs is almost negligible, the authors could comment on the potential harzards and the specific measures to consider when handling such reactions. Also, it seems appropriate to comment on how scalable is the IL synthesis (with yields below 60%) for being implemented an industrial level.”
We thank the reviewer for the comment. As the reviewer pointed out and we totally agreed that we hardly claimed in this work as a sensor for amine and alcohol gases, we only stated that our work was a “reaction-based amine and alcohol gases detection with triazine ionic liquid materials” (our manuscript title) and a “chemoselective detection of amine and alcohol gases by a quartz crystal microbalance” (presented in the Conclusion of manuscript). As also the reviewer pointed out and we agreed, in order to successfully synthesize AILs, our AIL synthesis was inevitably involved volatile solvents and hazardous chemicals. Our work presented, at this stage, was purely an academic research.
‘’-A proper characterization of the mechanical response of the resonator, i.e. quality factor and natural frequency, before and after the deposition of the ionic liquid should be provided in the manuscript. The modification of the QCM sensor with the ionic liquid is done by carefully addition with a pipette of the prepared IL solution to get 33 nm/300 nm of sensing coating. More information about the mass and thickness estimation (SEM analysis, AFM, perfilometer) is required. In addition, the viscosity of the IL, and the viscoelastic behaviour of the IL thin film should be discussed in the manuscript and also considered when applying the Sauerbrey equation to calculate the mass variation changes, i.e. IL loading, from the frequency shift.”
We thank the reviewer for the constructive comments. As the reviewer pointed out and we agreed, we have accordingly addressed the reviewer points by (i) including info on masses of AIL 1 and AIL 2 used in QCM measurements in the caption of Figure 2 (line 130, page 4) and (ii) relocating the section of ‘QCM measurements’ originally deposited in the Supporting Information to create a totally new section paragraphs (lines 324-345, pages 9 and 10), labeled in red, in the revised manuscript.
“-The authors underlies the chemoselective detection of gaseous amines and alcohols, but do not carry out detection experiments with multicomponent mixtures to confirm such selective detection.
-Experimental details about the preparation of N2 stream (3 mL/min) containing 100 ppb of the target analyte are missing.”
We thank the reviewer for constructive comment. As the reviewer pointed out and we agreed, we have accordingly addressed the reviewer point by relocating the section of ‘QCM measurements’ originally deposited in the Supporting Information to create a totally new section paragraphs (lines 324-345, pages 9 and 10) with detailed experimental sample preparation and QCM measurements, labeled in red, in the revised manuscript. Also, our work presented in this work, at this stage, was purely an academic research.
“-“Sensitivity” of detection, “Limit of Detection”, “Response Time” and “Recovery Time” parameters have to be properly defined and calculated from the experiments carried out. A zooming of the “y” axis of Figures 2 to 5 will be clearly beneficial to assess on the noise level. In addition, specific comments on the “reusability” of the sensors, i.e. reaction irresversibility, should be given in the text.
-Please avoid auto-cite in the introduction when describing the QCMs as gas sensors.”
We thank the reviewer for comment. In this work, we only used the term “sensitivity of detection” for the concentration of an analyte such as amine that produced a 10 Hz-decrease in resonance frequency (i.e., ΔF = -10 Hz) (lines 149-151, page 4). Also, the used quartz chips could readily be regenerated by washing with solvents (acetonitrile or methanol) and coating again the AILs (lines 339-340, page 9), labeled in red, in the revised manuscript.
We have also addressed the following points raised by the reviewer 3:
“The manuscript has been well organized, only several minor points need to be revised:
Title: Is correct “Gase detection”?
Abstract: Line 10. Please include here QCM abbreviation after the complete name.
Line 51: …group would be excellent…
Line 113: Include here the complete name for ESI-HRMS. The same for TMS in line 183.”
We thank the reviewer for constructive comments. As the reviewer pointed out and we totally agreed, we have accordingly addressed the reviewer points by (i) correcting the typo at manuscript title (line 2, page 1), (ii) including ‘QCM’ in Abstract (line 10, page 10), (iii) deleting ‘…be…’ and ‘…candidates…’ (line 52, page 2), and (iv) providing the full name for ESI-HRMS (line 123, page 4) and TMS (line 194, page 6), labeled in red, in the revised manuscript.
“Section 3.2. The abbreviation used in Scheme 1 would be very valuable here to easily follow the synthesis procedures. For example, Boc2O, MsCl, TFA …”
References: In my opinion two references should be included: Toniolo et al., Anal. Chem. 85, 2013, 7241-7247. Xu et al., Sensors and Actuators B 134, 2008, 258-265.”
We thank the reviewer for constructive comments. As the reviewer pointed out and we totally agreed, we have accordingly addressed the reviewer points by (i) providing the abbreviated chemicals their full names in the caption of Scheme 1 (lines 91-92, page 3), labeled in red, in the revised manuscript. Also, we agree to add the two references suggested by the reviewer (ref# 26 and 27, page 11), labeled in red, in the revised manuscript.
We have also addressed the following points raised by the reviewer 4:
“This manuscript reported the synthesis of two new ionic liquids and the application of them as chemoselective vapor sensors. The main concern is the sensing responses rely on the nucleophilic aromatic substitution, which gives selective sensing performances. However, on the other hand, this reaction would resulted in the irreversible sensing as the amines cannot be eliminated by purging with N2 gas. The ability of recovery is a key criterion in sensing technology. Therefore, I think the main claim of this manuscript is not acceptable. It is confusing of the statement of sensing of amine: “for isobutylamine gas (a bacterial volatile) at 10 Hz decrease in resonance frequency”. It is described in Figure 2, the response is -175 Hz. How is the sensitivity of detection of 6.3 ppb calculated? In terms of experimental details: 1) How was vapor concentration in ppb level realized in the experiments? How was it calibrated? 2) how were the mass spectrometry experiments performed/ How were the sample prepared, on QCM substrates or bulk synthesis?”
We thank the reviewer for constructive comments. As this and other reviewers pointed out the inquiry of the experimental details in QCM preparation and measurements and we totally agreed, we have accordingly addressed the reviewer points by relocating the section of ‘QCM measurements’ originally deposited in the Supporting Information to create and place a totally new section paragraphs (lines 324-345, pages 9 and 10), labeled in red, in the revised manuscript. Also, as clearly stated in the text (lines 149-151, page 4) as well as the caption of Figure 4 (line 168, page 6), the value of 6.3 ppb was obtained from 10 Hz / 1.5874 (= 6.3 ppb) in the revised manuscript.
We have revised our manuscript and addressed all points raised by all four reviewers. We hope that our manuscript is now in suitable form for publication.
Best wishes.
Very truly yours,
Yen-Ho Chu
Distinguished Professor of Chemistry and Biochemistry
National Chung Cheng University
Chiayi 62102, Taiwan, ROC
Phone: +886-5-272-0411 ext. 66408
Fax: +886-5-272-1040
E-mail: cheyhc@ccu.edu.tw

Round 2
Reviewer 2 Report
The following concerns have not been properly addresed in the revised version:
-The high toxicity of some of the chemicals used for the synthesis of triazine based ionic liquids: cyanuric chloride and methyl cyanide. Although the vapor pressure of the resulting ILs is almost negligible, the authors could comment on the potential hazards and the specific measures to consider when handling such reactants. Also, it seems appropriate to comment on how scalable is the IL synthesis (with yields below 60%) for being implemented at industrial level.
-A proper characterization of the mechanical response of the resonator, i.e. quality factor and natural frequency, before and after the deposition of the ionic liquid should be provided in the manuscript. More information about the thickness estimation (SEM analysis, AFM, perfilometer) is required. In addition, the viscosity of the IL, and the viscoelastic behaviour of the IL thin film should be discussed in the manuscript and also considered when applying the Sauerbrey equation to calculate the mass variation changes, i.e. IL loading, from the frequency shift.
-"Sensitivity" of detection, "Limit of Detection", "Response Time" and "Recovery Time" parameters have to be properly defined according to the IUPAC guidelines and calculated from the experiments carried out. A zooming of the "y" axis of Figures 2 to 5 will be clearly beneficial to asses on the noise level.
Reviewer 4 Report
Thanks for the authors’ revision. However, the comment on reversible sensing performance of these ionic liquid based QCM sensors was not addressed. As this sensing response strongly relies on the chemical reaction between the sensory materials and the analytes, it would be difficult to regenerate the sensors. I totally agree with the comments of referee 2 that the QCM platform is useful to study the reaction between ionic liquids and amines. But I do not think these ionic liquids are potential sensory materials for sensing of amines, at least not suitable in this configuration. In terms of the QCM results, I suggest the authors to re-organize this manuscript focusing on the understanding of chemical reactions instead of gas sensing.
